# Entropy Weighted Adversarial Training

**Minseon Kim** [1]   **Jihoon Tack** [1]   **Jinwoo Shin** [1]   **Sung Ju Hwang** [1 2]

## Abstract

Adversarial training methods, which minimizes the loss of adversarially-perturbed training examples, have been extensively studied as a solution to improve the robustness of the deep neural networks. However, most adversarial training methods treat all training examples equally, while each example may have a different impact on the model's robustness during the course of training. Recent works have exploited such unequal importance of adversarial samples to model's robustness, which has been shown to obtain high robustness against untargeted PGD attacks. However, we empirically observe that they make the feature spaces of adversarial samples across different classes overlap, and thus yield more high-entropy samples whose labels could be easily flipped. This makes them more vulnerable to targeted adversarial perturbations. Moreover, to address such limitations, we propose a simple yet effective weighting scheme, *Entropy-Weighted Adversarial Training* (EWAT), which weighs the loss for each adversarial training example proportionally to the entropy of its predicted distribution, to focus on examples whose labels are more uncertain. We validate our method on multiple benchmark datasets and show that it achieves an impressive increase of robust accuracy.

## 1. Introduction

The deep neural networks (DNN) often output incorrect predictions even with small perturbations to the input examples (Szegedy et al., 2013), despite their impressive performances on a variety of real-world applications. This adversarial vulnerability is a crucial problem in deploying them to safety-critical real-world applications, such as autonomous driving or medical diagnosis. To tackle the vulnerability,

*Equal contribution [1]Korea Advanced Institute of Science and Technology (KAIST), Daejeon, South Korea [2]AITRICS, Seoul, South Korea. Correspondence to: Sung Ju Hwang <sjhwang82@kaist.ac.kr>.

*Accepted by the ICML 2021 workshop on A Blessing in Disguise: The Prospects and Perils of Adversarial Machine Learning.* Copyright 2021 by the author(s).

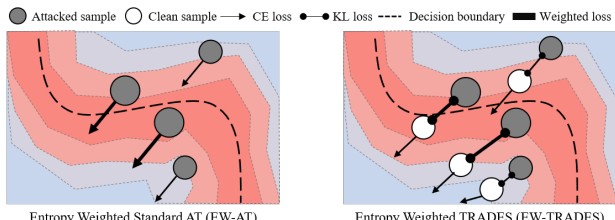

*Figure 1.* **Overview of our approach.** EWAT weighs more on the uncertain examples which have large entropy (red) while adjust relatively low weights on the low entropy examples (blue). For standard AT, weighting is applied on the cross-entropy loss. For TRADES, weighting is applied on the Kullback-Leibler loss.

various approaches have been proposed to ensure the robustness against adversarial attacks (Madry et al., 2018; Zhang et al., 2019; Kurakin et al., 2016; Wang et al., 2019).

The most promising approach to improve adversarial robustness of DNN is *adversarial training*, which trains the model to minimize the loss on the adversarially perturbed examples. Goodfellow et al. (2015) propose to train the model with samples attacked with Fast Gradient Sign Method (FGSM). Following this work, various adversarial defense algorithms have been suggested. For example, Adversarial Training (standard AT) (Madry et al., 2018) uses a min-max formulation where the examples are perturbed with the loss maximization objective with the Projected Gradient Descent (PGD) attack. Further, TRADES (Zhang et al., 2019) demonstrates the trade-off between the clean accuracy and robustness, and proposes to minimize the Kullback-Leibler divergence (KL) between the prediction on the clean example and its adversarial counterpart, to achieve robustness against adversarial perturbations.

In natural image classification training, some works have shown that only a small portion of examples from the training set contribute to the generalization performance (Toneva et al., 2019), where each sample has a different impact on the model's final performance. Similarly, it is also natural to assume that some training examples are more important than others, in enhancing the adversarial robustness of the adversarially trained model.

Similar to this intuition, recent studies suggest to identify such robustness-critical instances, to assign more weights on them during adversarial training. To name a few, Wang et al. (2019) argue that misclassified *clean* samples are

more important in achieving robustness, and imposes larger KL regularization on them (MART). On the other hand, Zhang et al. (2020) suggest to assign more weights on examples that were close to the decision boundary before the adversarial attack (GAIRAT). These methods have shown to achieve impressive robustness against untargeted PGD attacks.

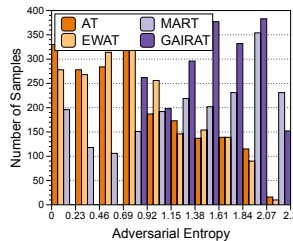

*Figure 2.* Distribution of the entropy for the adversarial samples obtained with different weighting methods.

However, we discover that weighting scheme in MART and GAIRAT create a false sense of robustness. In short, while they appear more robust against untargeted PGD attacks, they become more vulnerable to other types of adversarial attacks, such as logit scaling attack (Hitaj et al., 2021) and AutoAttack (Croce & Hein, 2020a), compared to standard AT. We further show that these weighting schemes make the feature spaces of adversarial samples belonging to different classes overlap (Figure 3), and thus increase the entropy of the adversarially-perturbed training examples (see Figure 2).

Such high-entropy samples are more vulnerable to targeted adversarial attacks, since their predicted labels are uncertain, and could be flipped with less efforts. Based on this observation, we propose a simple yet effective weighted adversarial training method that improves the model's robustness, which assigns a weight to each adversarially-perturbed sample based on the entropy of its predicted distribution. Specifically, our method assigns larger weights to training examples with high entropies (see Figure 1).

The experimental validation of our weighted adversarial training scheme, named Entropy-Weighted Adversarial Training (EWAT), verifies that it improves the robustness of the existing adversarially-trained models. Our weighting scheme is simple to implement, compute, and use, while improving the robustness without any additional computational cost. In summary, the contributions are as follow:

- We empirically observe that the previous weighting schemes make the feature spaces of adversarial samples across different classes overlap, and thus yield more high-entropy samples whose labels could be easily flipped which induce vulnerability against the AutoAttack and logit scaling attack.

- Based on the observation, we propose a surprisingly simple, yet effective **entropy weighting** scheme that can enhance model's robustness, which weighs the loss of adversarial samples with respect to their entropy. Our approach improves the robustness of the model against AutoAttack and logit scaling attack, to which

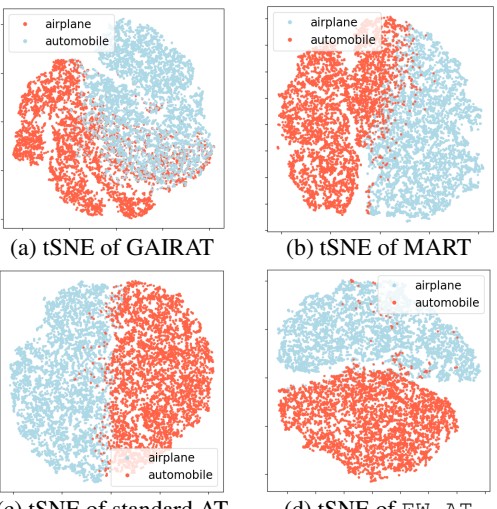

| (a) tSNE of GAIRAT | (b) tSNE of MART |
| (c) tSNE of standard AT | (d) tSNE of EW-AT |

*Figure 3.* Visualization of the embeddings of adversarial examples from each model. All models are trained with PreActResNet18. We only visualize train examples from two classes (airplane and automobile) in CIFAR10 dataset.

the existing weighting models are vulnerable.

## 2. Unequal Importance of Each Sample in Adversarial Training

In this section, we elaborate on what should be considered in instance-wise weighted adversarial training, in order to consider the unequal importance of each sample to the adversarial robustness of the model. Moreover, we also show the adversarial vulnerability of the previous instance weighting schemes for adversarial training. To be precise, this vulnerability does not come from any types of the obfuscated gradients introduced in the Athalye et al. (2018), but it also creates a false sense of robustness.

To design the weighting scheme in adversarial training, we first have to define two conditions.

*1) Which criteria should we use to evaluate the importance of samples during adversarial training?*

*2) How can we assign attention/weight to differently contributed samples?*

The previous works answered both questions with their intuitions, and verified their hypotheses with the empirical results on the untargeted PGD attacks. MART (Wang et al., 2019) argues that the predictions on the non-perturbed samples are important criteria to measure the sample-wise importance in adversarial training. Thus, MART assigns more adversarial attentions to KL loss that have low confidence before perturbation (Eq (7)). GAIRAT (Zhang et al., 2020), on the other hand, hypothesizes that the number of steps to perturb the given sample is an important measure of its importance in adversarial training. Thus, GAIRAT assigns

*Table 1.* Vulnerability loophole in the previous weighted adversarial training methods. We validate MART and GAIRAT against logit scaling attack (LS) (Hitaj et al., 2021) with $\alpha$=10 and the AutoAttack (AA) (Croce & Hein, 2020a) with $\epsilon = 0.031$. All models are trained with PreActResNet18 architecture.

| Method | PGD | LS | AA |
|---|---|---|---|
| GAIRAT (Zhang et al., 2020) | 55.16 | 31.78 | 22.37 |
| MART (Wang et al., 2019) | **57.08** | 48.70 | 46.79 |
| standard AT (Madry et al., 2018) | 53.96 | 51.26 | 48.16 |
| + Ours (`EW-AT`) | 53.49 | 51.81 | 49.20 |
| TRADES (Zhang et al., 2019) | 53.95 | 50.10 | 49.30 |
| + Ours (`EW-TRADES`) | 53.83 | 50.13 | **49.90** |

more attentions on the adversarial samples that violate the margin more (Eq (8)). (Equations are in Appendix A)

However, we empirically find that the performance achieved by weighting only improves untargeted PGD attacks. Yet, both methods achieve lower performance compared to standard AT against logit scaling attack (Hitaj et al., 2021) and AutoAttack (Croce & Hein, 2020a) (see Table 1). We further examine why the previous weighting schemes are vulnerable to non-PGD attacks, by visualizing the t-SNE embeddings of the adversarially perturbed training samples in Figure 3. As shown in Figure 3, adversarial examples generated by GAIRAT and MART for two different classes have large overlaps, while the t-SNE of the adversarial samples trained with standard AT shows clear separation.

This is because the previous weighting schemes make the model only focus on certain samples that they deem difficult, making the prediction on others more uncertain. This will make the entropy of the predictive distribution of such samples to be high. This is shown in Figure 4a, where MART and GAIRAT have more than double the number of 'high-entropy' samples ($> 1.5$). It is evident that such high-entropy samples will be more prone to make wrong predictions if they are perturbed only a little to a manifold of another class that is predicted high, thus making them more vulnerable to targeted attacks. This suggests that if we can minimize the existence of such high-entropy samples, the model's robustness will be improved. We propose our method which exploits this observation in the next section.

## 3. Entropy-Weighted Adversarial Training

We now describe our method, Entropy-Weighted Adversarial Training, which weighs the loss of each adversarial example based on its entropy of the predictive distribution.

In the previous section, we showed that GAIRAT and MART make many of its adversarial samples to have relatively high entropies compared to adversarial samples from standard AT, as shown in Figure 2, and that this makes them vulnerable against AutoAttack (Croce & Hein, 2020a) (Figure 4a). They have created more vulnerable samples while trying to focus on samples they deem as important, by assigning

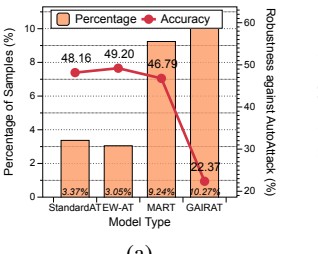 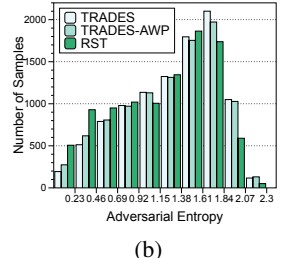

(a)         (b)

*Figure 4.* (a) Percentage of high entropy samples (>1.5) in test set from standard AT, `EW-AT`, MART, and GAIRAT which shows correlation to performance of AutoAttack. (b) Distribution of adversarial entropy in test set from TRADES, TRADES-AWP, and RST which utilize same TRADES loss during the training.

relatively smaller weights on other samples, making the prediction confidence on them low.

Further, we also empirically observe that models that are robust against AutoAttack (TRADES < TRADES-AWP < RST) have relatively smaller ratio of high entropy examples (Figure 4b). We further compare the model that utilizes same form of $\mathcal{L}_{KL}$ loss in Figure 4b. Notably, a model that is more robust against AutoAttack can easily classify the samples that are large entropy. All of these empirical evidences suggest that the ratio of high entropy samples is highly related to the model's robustness. Also, the entropy is a more direct measure of a sample's robustness, unlike its distance to the margin (GARIAT) or whether the sample is predicted incorrectly (MART), since a high entropy sample's predicted label could be altered more easily. Thus, we propose to consider the *entropy* of each adversarially perturbed sample as a criterion to measure its vulnerability, and propose a loss weighting scheme based on the entropy.

**Entropy.** The *entropy* $\mathcal{E}$ is a measurement of state of the uncertainty, and randomness. The entropy for an adversarial sample $x^{\mathrm{adv}}$ for classification tasks can be defined as follows:

$$\mathcal{E}(\theta, x) = - \sum_{j=1}^{C} p_j(f(\theta, x)/\tau) \log \left( p_j(f(\theta, x)/\tau) \right), \quad (1)$$

where $p_j$ stands for the $j^{\mathrm{th}}$ class softmax probability of $f(\cdot)$, C is the number of classes and $\tau$ is temperature scaling factor where we set as 1. We can control $\tau$ to affect the $\mathcal{E}$ by making the predictive distribution to be sharper or smoother.

**Entropy-based sample weighting.** We now propose an additional entropy-weighted loss term for adversarial training, which weighs each adversarial example by its entropy.

The entropy value of each training example continuously changes during the course of training. This is beneficial since the weighting changes adaptively, such that it focuses on the most uncertain samples at each iteration. However, one caveat here is that entropies of all samples will go low as the model trains on, which will simply have small or no effects of weighting. Since this will be the same as

*Table 2.* **Results against $\ell_\infty$ attack in CIFAR10 with WideResNet34-10.** Clean denotes the accuracy on natural images. Best and Last stand for the best robust accuracy, and the accuracy at the last epoch, against PGD10, respectively. The robust accuracy against PGD is calculated against $\epsilon = 8/255$. For the AutoAttack, we use the threat model with $\epsilon = 0.031$. The bold fonts indicate that cases where EWAT improves the performance of the base adversarial training method.

| | Last | | Best | | |
|---|---|---|---|---|---|
| Method | Clean | PGD10 | Clean | PGD100 | AutoAttack (worst attack) |
| GAIRAT (Zhang et al., 2020) | 85.24 | 52.97 | 86.16 | 57.37 | 42.28 |
| MART (Wang et al., 2019) | 83.72 | 55.73 | 82.85 | 59.30 | 51.39 |
| standard AT (Madry et al., 2018) | 87.38 | 54.21 | 85.84 | 56.17 | 52.07 |
| + Ours (EW-AT) | 86.97 | 54.69 | 85.39 | 55.54 | **52.51 (+0.44)** |
| TRADES (Zhang et al., 2019) | 85.62 | 57.32 | 85.62 | 57.54 | 53.82 |
| + Ours (EW-TRADES) | 83.11 | 57.84 | 82.54 | 58.27 | **54.58 (+0.76)** |

non-weighted training at the end, we normalize the entropy weights with the batch-mean of the entropy at each iteration. Formally, for a given batch of adversarial examples $\mathcal{B} := \{(x_i^{\mathrm{adv}}, y_i)\}_{i=1}^m$, we define the entropy weighting ($w_i^{\mathrm{ent}}$) for a given instance $x_i^{\mathrm{adv}}$ as follow:

$$w_i^{\mathrm{ent}} = \frac{1}{\eta} \cdot \mathcal{E}(\theta, x_i^{\mathrm{adv}}), \qquad (2)$$

where $\eta := \sum_{x^{\mathrm{adv}} \in \mathcal{B}} \mathcal{E}(\theta, x^{\mathrm{adv}})/B$ is the batch mean of the predicted entropy. The final objective consisting of the original adversarial training loss and entropy weighted cross-entropy loss is as follows:

$$\mathcal{L}_{\mathrm{Entropy}} = w^{\mathrm{ent}} \cdot \mathcal{L}_{\mathrm{CE}}\big(f(\theta, x^{\mathrm{adv}}), y\big),$$
$$\mathcal{L}_{\mathrm{EW\text{-}AT}} = \mathcal{L}_{\mathrm{AT}} + \mathcal{L}_{\mathrm{Entropy}} \qquad (3)$$
$$= (1 + w^{\mathrm{ent}}) \cdot \mathcal{L}_{\mathrm{CE}}\big(f(\theta, x^{\mathrm{adv}}), y\big).$$

(See Appendix C.1 for TRADES loss.)

# 4. Experiments

**Against standard attacks.** Our entropy weighted adversarial training improves upon the baselines, outperforming standard AT model by 0.44%, and TRADES by 0.76%, with the WideResNet34-10 model (Table 2) against AutoAttack. Considering that TRADES is considered as powerful by making 1% improvement over the standard AT, this is a meaningful improvement of the robust accuracy. On the contrary, GAIRAT and MART achieve lower robustness over standard AT against AutoAttack, although they attempted to improve upon the standard AT model by the proposed weighing schemes.

In practice, we cannot assume that the attacker will only use a single type of attack, and thus the most important measures of robustness is the robust accuracy against the strongest attack, which is the AutoAttack in this case. Our EWAT shows high robustness against this worst-case attack, although it also obtains comparable performance to baselines', against the PGD attacks.

*Table 3.* **Results against $\ell_\infty$ attack in MNIST, and CIFAR100.** Clean denotes the accuracy of the natural images at the last epoch. For the AutoAttack, we use the threat model with $\epsilon = 0.031$.

| | MNIST | | CIFAR100 | |
|---|---|---|---|---|
| Method | Clean | AutoAttack | Clean | AutoAttack |
| standard AT (Madry et al., 2018) | 98.74 | **88.51** | 55.72 | 24.09 |
| + Ours (EW-AT) | 98.97 | 88.43 (-0.08) | 57.71 | **24.57 (+0.48)** |
| TRADES (Zhang et al., 2019) | 98.37 | 89.30 | 57.78 | 25.06 |
| + Ours (EW-TRADES) | 97.46 | **89.71 (+0.41)** | 55.42 | **25.66 (+0.60)** |

**Against logit scaling attack and AutoAttack.** Previous weighting methods, MART and GAIRAT, suffer from the low robustness against logit scaling attack (Hitaj et al., 2021) and AutoAttack. However, our model demonstrates improved robustness against both types of attacks (Table 1) and does not suffer from the vulnerability loophole, unlike the existing loss weighting schemes.

**Results on multiple benchmarks datasets.** We validate our methods on multiple benchmarks datasets. In Table 3, EWAT consistently improve upon standard AT and TRADES against AutoAttack on MNIST (LeCun et al., 1990), and CIFAR100 (Krizhevsky et al., 2009). Compared to the margin-based methods, our model does not require any warm-up epochs for weighting instances, even on larger datasets such as CIFAR100. This is because it relies on entropy, which can be computed easily and is well defined regardless of the training stage, unlike other values, such as distance to the (estimated) margins. Moreover, our methods work better on a larger dataset (CIFAR100) with more number of classes, on which the model's predictions could be more uncertain, due to the increased confusion across the classes, than on a smaller dataset (MNIST) with few classes.

# 5. Discussion

In this paper, we showed that existing weighting schemes for adversarial training yield high-entropy examples with uncertain predictions, thus making them vulnerable to targeted attacks such as AutoAttack. Based on this observation, and the direct association of the entropy to its vulnerability to targeted attacks, we propose *Entropy-Weighted Adversarial Training* (EWAT), which is a simple yet effective weighted adversarial training scheme which to weigh each instance by its entropy, EWAT is simple to implement and incurs no additional cost, and can be used to weigh the instance-wise adversarial loss of any conventional adversarial training algorithms, such as standard AT and TRADES. Moreover, while existing instance weighting scheme for adversarial training suffer from vulnerability against logit scaling attack and AutoAttack, our weighting scheme achieves better robustness against them over standard AT and TRADES with even weights across the samples. Further, it also achieves competitively robust accuracy against untargeted PGD attack to standadrd AT and TRADES. The experimental results on multiple benchmarks datasets further demonstrates the versatility and generalizability of our method.

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

# Appendix

## Entropy Weighted Adversarial Training

## A. Related Work

**Adversarial robustness.** Szegedy et al. (2013) firstly showed that deep neural networks for image classification are vulnerable to imperceptible *small perturbations* applied to input images. To achieve robustness against such adversarial attacks, Goodfellow et al. (2015) proposed Fast Gradient Sign Method (FGSM), which perturbs a target sample to its gradient direction to increase its loss. Then, they proposed an adversarial training objective which aim to minimize the loss on the perturbed samples as well as clean samples, which have shown to be effective against such adversarial attacks. Follow-up works (Moosavi-Dezfooli et al., 2016; Kurakin et al., 2016; Carlini & Wagner, 2017) proposed a variety of gradient attacks that are stronger than FGSM that can be used for adversarial training, and Madry et al. (2018) proposed a minimax formulation to minimize the loss of adversarial examples, which are perturbed to maximize its loss with projected gradient method. After a surge of interest in adversarial robustness of neural networks, various defense mechanisms (Song et al., 2017; Buckman et al., 2018; Dhillon et al., 2018) have been proposed to defend against such adversarial attacks. However, Athalye et al. (2018) showed that many of them except *standard AT*, rely on gradient masking, which results in obfuscated gradient in the representation space, and are highly vulnerable to stronger attacks that circumvent it. TRADES (Zhang et al., 2019) propose to minimize the Kullback-Leibler divergence (KL) between a clean example and its adversarial counterpart, to enforce consistency between their predictions, and further showed that there is a theoretical trade-off between clean accuracy of a model and its robustness. Recently, using additional unlabeled data (Carmon et al., 2019) or using additional attack mechanism (Wu et al., 2020) have been proposed. To utilize the additional data, Carmon et al. (2019) propose to use Tiny ImageNet (Le & Yang, 2015) as pseudo-label to learn more rich representation of CIFAR10 (Krizhevsky et al., 2009) dataset that could lead to robust model (RST). Wu et al. (2020) propose a double-perturbation mechanism which conducts additional adversarial weight perturbation (AWP) with conventional adversarial training.

**Instance-wise weighting for adversarial training.** While successful in general, none of aforementioned works consider the varying impact of samples on adversarial robustness. A recent work, Misclassification Aware adveRsarial Training (MART) (Wang et al., 2019), focuses on this problem and propose to put more weights on the misclassified clean samples for the KL-divergence regularization, achieving state-of-the-art robustness against untargeted PGD attacks. Furthermore, another recent work, Geometry-aware Instance-Reweighted Adversarial Training (GAIRAT) (Zhang et al., 2020) proposed a method with a similar motivation, which weighs the adversarial loss of each sample based on the clean sample's distance to decision boundary. GAIRAT also achieves impressive performance against untargeted PGD attacks. However, these methods make increase the entropies of the adversarially perturbed samples, and thus make the model to be more vulnerable against targeted attacks, such as AutoAttack (Croce & Hein, 2020a) and the logit scaling attack (Hitaj et al., 2021). We observe that both instance weighting methods for the adversarial training largely increase the entropies of the perturbed examples, which make the samples more vulnerable as their predictions are easier to alter.

**Adversarial attacks.** Most of the adversarial defense mechanisms have been broken with stronger attacks which were not aware of at the time they were first introduced. Athalye et al. (2018) is an important work that has helped many researcher to explore means to achieve fundamental robustness rather than take advantage of false sense of security created with the obfuscated gradients. Recently, Croce & Hein (2020a) proposed an ensemble attack that consists of four different attacks (AutoAttack), namely untargeted APGD-CE, targeted APGD-DLR, FAB (Croce & Hein, 2020b) and square attack (Andriushchenko et al., 2020). APGD-CE and APGD-DLR are step size-free variants of the PGD attack. AutoAttack revealed that the most of defense methods are actually more vulnerable than TRADES, if the attacker carries out a targeted attack, which is a more viable scenario in the real-world cases.

### A.1. Preliminaries

In this section, we first recap the adversarial training (standard AT) (Madry et al., 2018), TRADES (Zhang et al., 2019) and previous instance weighting methods for adversarial training (MART (Wang et al., 2019), GAIRAT (Zhang et al., 2020)).

Let us denote the dataset $\mathcal{D} = \{X, Y\}$, where $x \in X$ is a training example and $y \in Y$ is its corresponding label, and a

supervised learning model $f_\theta : X \to Y$ where $\theta$ is the set of the parameters of the model. Given such a dataset and a model, *adversarial attacks* aim towards finding the worst-case examples, by searching for the perturbation for each example that maximizes the loss within a certain radius from it (e.g., norm balls). We can define such adversarial $\ell_\infty$ attacks as follows:

$$\delta^{t+1} = \Pi_{B(0,\epsilon)}\Big(\delta^t + \alpha\,\mathtt{sign}\big(\nabla_{\delta^t}\mathcal{L}_{\mathrm{CE}}\big(f(\theta, x + \delta^t), y\big)\big)\Big), \tag{4}$$

where $B(0, \epsilon)$ is the $\ell_\infty$ norm-ball with radius $\epsilon$, $\Pi$ is the projection function to the norm-ball, $\alpha$ is the step size of the attacks and $\mathtt{sign}(\cdot)$ is the sign of the vector. Further, the perturbation $\delta$ is the accumulated $\alpha\,\mathtt{sign}(\cdot)$ over multiple attack iterations $t$, and $\mathcal{L}_{\mathrm{CE}}$ is the cross-entropy loss. In case of Projected Gradient Descent (PGD) (Madry et al., 2018), the attack starts from a random point within the epsilon ball and performs $t$ gradient steps, to obtain a perturbed sample $x^{\mathtt{adv}}$.

The most straightforward way to defend against such adversarial attacks is to minimize the loss of adversarial examples, which is often called *adversarial training*. The standard AT framework proposed by Madry et al. (2018) solves the following min-max problem where $\delta$ is the perturbation of the adversarial example of the given input $x$, and $y$ is its target class label. Then the loss is:

$$\mathcal{L}_{\mathrm{AT}} = \max_{\delta \in B(x,\epsilon)} \mathcal{L}_{\mathrm{CE}}\big(f(\theta, x + \delta), y\big). \tag{5}$$

Another popular algorithm for adversarial training, TRADES (Zhang et al., 2019), suggests to minimize the Kullback-Leibler (KL) divergence between a clean example and its adversarial perturbation, to enforce consistency between their predictions while using cross-entropy loss on clean samples as follow:

$$\mathcal{L}_{\mathrm{TRADES}} = \mathcal{L}_{\mathrm{CE}}\big(f(\theta, x), y\big) + \beta \max_{\delta \in B(x,\epsilon)} \mathcal{L}_{\mathrm{KL}}\big(f(\theta, x)||f(\theta, x + \delta)\big), \tag{6}$$

where $\mathcal{L}_{\mathrm{KL}}$ is KL divergence loss and $\beta$ is a parameter to control the trade-off between clean accuracy and adversarial performance.

**Instance weighting schemes for adversarial training.** Recently, MART (Wang et al., 2019) proposed a new weighted adversarial training framework with the boosted cross entropy loss and the weighted KL divergence loss. The boosted cross entropy loss maximizes the $1-$ the second highest class probability, to increase the margin of the classifier. The weighted KL divergence loss assigns higher weights on the KL-divergence term, for the samples that are misclassified before applying the adversarial perturbations. The loss of MART is defined as follows:

$$\mathcal{L}_{\mathrm{MART}} = \mathcal{L}_{\mathrm{AT}} - \log\big(1 - \max_{k \neq y} p_k(f(\theta, x + \delta))\big) + \lambda \mathcal{L}_{\mathrm{KL}}\big(f(\theta, x)||f(\theta, x + \delta)\big)\big(1 - p_y(f(\theta, x))\big), \tag{7}$$

where $p_k$ is the probability of $k^{th}$ class.

A recent approach, GAIRAT (Zhang et al., 2020), suggests a loss weighting scheme based on the clean sample's distance to the decision boundary:

$$\mathcal{L}_{\mathrm{GAIRAT}} = \frac{\gamma}{\sum_{i=0}^{B} \gamma} \mathcal{L}_{\mathrm{AT}},$$

$$\gamma = \frac{\big(1 + \mathtt{tanh}\big(\psi + 5(1 - 2\kappa(x, y)/K)\big)\big)}{2}, \tag{8}$$

where $B$ is batch size, and $\kappa(x, y)$ is geometric distance of a data point $(x, y)$. $\kappa(x, y)$ is calculated as total number of attack steps minus least number of necessary attacked-steps to change the label $y$ of $x$ during the PGD attack Eq. (4). $\psi$ is a constant hyperparameter and $K$ is total attack steps. Therefore, if the sample is already far from the decision boundary, those samples are not used during the training. This causes the highly underconfident model and induces vulnerability against AutoAttack (Croce & Hein, 2020a) and logit scaling attack (Hitaj et al., 2021).

# B. Detailed description of experimental setups

### B.1. Resource description.

All experiments are conducted with a single GPU (NVIDIA RTX 2080 Ti), except for the TRADES experiments with WideResNet in Table 2. For WideResNet TRADES, two GPUs (NVIDIA RTX 2080 Ti) are used. All experiments are processed in Intel(R) Xeon(R) Silver 4114 CPU @ 2.20GHz.

## B.2. Dataset description.

For experiments, we use CIFAR 10, CIFAR 100, and MNIST. CIFAR 10 and CIFAR 100[1] consist of 50,000 training images and 10,000 test images with 10 and 100 classes, respectively. All CIFAR images are $32\times32\times3$ resolution (width, height, and channel). MNIST dataset contains hand-written digits, ranging from 0 to 9. MNIST contains a training set of 60,000 examples and a test set of 10,000 examples, where each image has $28\times28\times1$ resolution (width, height, and channel).

## B.3. Training detail.

**MNIST.** For all methods compared, we train the network with $\ell_\infty$ attacks with the attack strength of $\epsilon = 0.3$ and the step size of $\alpha = 0.01$, with the number of inner maximization iteration set to $K = 40$. For optimization, we train every model for 100 epochs using the SGD optimizer with the weight decay of $1e{-}4$ and the momentum of 0.9. As for learning rate scheduling, we use the decay of 0.1 at the $20^{th}$ and $40^{th}$ epoch with the initial learning rate of 0.01.

**CIFAR.** For all methods, we train the network with $\ell_\infty$ attacks with the attack strength of $\epsilon = 8/255$ and the step size of $\alpha = 2/255$, with the number of inner maximization iteration set to $K = 10$. For the optimization, we train every model for 100 epochs using the SGD optimizer with the weight decay of $5e{-}4$ and the momentum of 0.9. For learning rate scheduling, we use the decay of 0.1 at the $100^{th}$ and $105^{th}$ epoch with the initial learning rate of 0.1.

**Hyperparameters.** When setting the hyperparameters for baselines, we follow their official settings in the original papers. For TRADES (Zhang et al., 2019), we set $\beta$ as 6.0, and for EW-TRADES we set $\beta$ as 5.5. In MART (Wang et al., 2019), we set $\lambda$ as 6.0. In GAIRAT (Zhang et al., 2020), we set $\psi$ as $-1.0$.

## B.4. Evaluation detail.

$\ell_\infty$ **attack.** For all $\ell_\infty$ attacks used in the test phase, we use the attack strength of $\epsilon = 8/255$ and the step size of $\alpha = 2/255$ with the number of inner maximization iteration set to $K = 10$ for PGD10. For PGD20 We use the $\alpha = \epsilon/10$ with $K = 20$, respectively.

**Logit scaling attack and AutoAttack.** We further test our EWAT against different type of attacks, e.g., Logit scaling attack (Hitaj et al., 2021) and AutoAttack (AA) (Croce & Hein, 2020a). Logit scaling attack is multiplying constant in the logit with $\alpha$ as follow:

$$\delta^{t+1} = \Pi_{B(0,\epsilon)}\Big(\delta^t + p\,\texttt{sign}\Big(\nabla_{\delta^t}\mathcal{L}_{\text{CE}}\big(\alpha \times f(\theta, x + \delta^t), y\big)\Big)\Big), \tag{9}$$

where $B(0, \epsilon)$ is the $\ell_\infty$ norm-ball with radius $\epsilon$, $\Pi$ is the projection function to the norm-ball, p is the step size of the attacks and $\texttt{sign}(\cdot)$ is the sign of the vector. We set $\alpha$ as 10 for testing in Table 1. When $\alpha$ is 1, logit scaling attack is the same as PGD attack. AutoAttack is an ensemble attack that is consists of four different attacks (APGD-CE, APGD-T, FAB-T (Croce & Hein, 2020b), and Square (Andriushchenko et al., 2020)). APGD-T and FAB-T are targeted attacks and Square is a black box attack[2].

# C. Entropy weighted Adversarial Training

## C.1. Entropy-based sample weighting for TRADES.

We now here describe an additional entropy-weighted loss term for TRADES, which weighs each adversarial example by its entropy on the KL term.

We will recap the entropy weight in here. Formally, for a given batch of adversarial examples $\mathcal{B} := \{(x_i^{\text{adv}}, y_i)\}_{i=1}^m$, we define the entropy weighting $(w_i^{\text{ent}})$ for a given instance $x_i^{\text{adv}}$ as follow:

$$w_i^{\text{ent}} = \frac{1}{\eta} \cdot \mathcal{E}(\theta, x_i^{\text{adv}}), \tag{10}$$

where $\eta := \sum_{x^{\text{adv}} \in \mathcal{B}} \mathcal{E}(\theta, x^{\text{adv}})/B$ is the batch mean of the predicted entropy.

---

[1]The full dataset of CIFAR can be downloaded at http://www.cs.toronto.edu/ kriz/cifar.html.
[2]AutoAttack https://github.com/fra31/auto-attack

For the TRADES loss, the overall weighted loss is as follows:

$$\mathcal{L}_{\texttt{Entropy}} = w^{\text{ent}} \cdot \mathcal{L}_{\text{KL}}\big(f(\theta, x) || f(\theta, x^{\text{adv}})\big),$$
$$\mathcal{L}_{\texttt{EW-TRADES}} = \mathcal{L}_{\texttt{TRADES}} + \mathcal{L}_{\texttt{Entropy}} \tag{11}$$
$$= \mathcal{L}_{\text{CE}}\big(f(\theta, x), y\big) + (\beta + w^{\text{ent}}) \cdot \mathcal{L}_{\text{KL}}\big(f(\theta, x) || f(\theta, x^{\text{adv}})\big).$$

## C.2. Algorithm.

---

**Algorithm 1** Entropy weighted adversarial training for (Madry et al., 2018)

---

**Input:** Dataset $\mathcal{D}$, parameters of model $\theta$, model $f$, number of epochs T, batch size m, number of batches M, Cross-entropy loss $\mathcal{L}_{\text{CE}}$, number of classes C

**for** epoch = 1, $\cdots$, T **do**

   **for** mini-batch = 1, $\cdots$, M **do**

      Sample mini-batch from training set ($\mathcal{D}$): $\{(x_i, y_i)\}_{i=1}^{m}$

      Generate adversarial examples $x_i^{\text{adv}}$ by Eq. 4

      Calculate entropy $\mathcal{E}$ for weighting

        $\mathcal{E}(\theta, x_i^{\text{adv}}) = -\sum_j^{\text{C}} p_j(f(\theta, x_i^{\text{adv}})) \log(p_j(f(\theta, x_i^{\text{adv}})))$

        $\eta = \frac{1}{m} \sum_{i=1}^{m} \mathcal{E}(\theta, x_i^{\text{adv}})$

        $w_{\texttt{ent}}^i = \mathcal{E}(\theta, x_i^{\text{adv}})/\eta$

      Calculate total loss

        $\mathcal{L}_{\texttt{EW-AT}} = \mathcal{L}_{\texttt{AT}} + w_{\texttt{ent}} \cdot \mathcal{L}_{\text{CE}}(f(\theta, x^{\text{adv}}), y)$

      Take gradient descent with respect to the model parameters

   **end for**

**end for**

---

# D. More experimental results

## D.1. Effect of temperature scaling.

We further examine the effects of simple temperature scaling, which is the only hyperparameter EWAT has (and is set to 1 by default), as it affects the entropy by making the predictive distribution sharper or smoother. We report the effect of different temperature values on our method's robust accuracy (Table 4). We observe that increasing the temperature value, which increases the overall entropy of all samples, improves the performance of EWAT.

*Table 4.* **Temperature scaling.** The reported results are robust accuracies against the $\ell_\infty$-AutoAttack on CIFAR10. $\tau$ is parameter for temperature scaling.

| Ours (EW-AT) | AutoAttack |
|---|---|
| $\tau = 0.5$ | 48.85 (+0.69) |
| $\tau = 1.0$ | 49.20 (+1.04) |
| $\tau = 5.0$ | 49.31 (+1.14) |

## D.2. Results of utilizing additional data.

Recently, adversarially training neural networks with generated images has shown to help with their robustness. Rebuffi et al. (2021) use samples generated by a Denoising Diffusion Probabilistic Model (DDPM; (Song et al., 2021)) to improve robustness. We utilize a dataset of 1M generated samples for CIFAR10 [3]. We train the model with WideResNet28-10 (Zagoruyko & Komodakis, 2016) with 55 epochs, learning rate decay at 45 and 50 epoch with 0.1.

Compare to RST, this method utilizes an in-distributed additional dataset which is more suitable for entropy-weighted. The dataset that is used in RST is from TinyImageNet which are out-of-distribution examples. Thus, entropy-weighted RST may behave in an undesirable way during the entropy-weighted training. For example, the model can have relatively high entropy to out-of-distributed images even though examples are used to train the

*Table 5.* **Results of using generated data against AutoAttack in CIFAR10.** The robustness accuracy is calculated with $\epsilon = 0.031$.

| Method | Clean | AutoAttack |
|---|---|---|
| DDPM (standard AT) | 84.99 | 53.89 |
| + Ours (EW-AT) | 84.93 | 54.02 (+0.13) |

in-distributed train set. However, DDPM is a generated model which generates additional images that have the same visual feature as the training set. Therefore, generated examples from DDPM can be seen as in-distributed examples. We believe that this is why our model can obtain improved robustness with DDPM generated images against AutoAttack differently from entropy-weighted RST, as shown in Table 5.

---

[3]The generated dataset can be downloaded at https://github.com/deepmind/deepmind-research/tree/master/adversarial_robustness