# OpenReview forum: "Entropy Weighted Adversarial Training"
_ICML.cc/2021/Workshop/AML — ICML 2021 Workshop AML Poster_

### Official Review · Reviewer_sJLZ · 2021-06-19
**The paper proposed Entropy-Weighted Adversarial Training(EWAT), which weighs the loss of each adversarial example based on its entropy of the predictive distribution.**

**Rating:** Accept
**Confidence:** 3

**Review:**

The paper considered the unequal importance of each sample in Adversarial Training(AT) and pointed out that related works such as MART and GAIRAT are vulnerable to target attack(e.g. AutoAttack) although they achieve impressive robustness against untargeted PGD attacks. Noticing high-entropy samples are more vulnerable to targeted adversarial attacks, the paper proposed Entropy-Weighted Adversarial Training(EWAT), which assigns a weight to each adversarially-perturbed sample based on the entropy of its predicted distribution. The experimental results show EWAT can improve the robustness against AutoAttack. The observation is interesting and the method is novel. However, EWAT isn’t as robust as MART against untargeted PGD attack(Table 2). What’s more, it’s better to evaluate EWAT against other targeted adversarial attacks

---

### Decision · Program_Chairs · 2021-06-21

**Decision:**

Accept (Poster)

**Comment:**

This paper proposed entropy-weighted adversarial training. The experiments show the effectiveness.